# Influence of Sex on the Microbiota of the Human Face

**DOI:** 10.3390/microorganisms10122470

**Published:** 2022-12-14

**Authors:** Clémence Robert, Federica Cascella, Marta Mellai, Nadia Barizzone, Flavio Mignone, Nadia Massa, Vincenzo Nobile, Elisa Bona

**Affiliations:** 1R&D Department, Complife Italia c/a Centre for Autoimmune and Allergic Diseases (CAAD), 22100 Novara, Italy; 2Centre for Autoimmune and Allergic Diseases (CAAD), University of Eastern Piedmont, 28100 Novara, Italy; 3Department of Health Sciences, University of Eastern Piedmont, 28100 Novara, Italy; 4Department of Science and Technologic Innovation, University of Eastern Piedmont, 15121 Alessandria, Italy; 5SmartSeq s.r.l., 28100 Novara, Italy; 6Department for Sustainable Development and Ecological Transition, University of Eastern Piedmont, 13100 Vercelli, Italy

**Keywords:** cutaneous microbiome, human face, 16S rDNA, gene sequencing

## Abstract

The role of the microbiota in health and disease has long been recognized and, so far, the cutaneous microbiota in humans has been widely investigated. The research regarded mainly the microbiota variations between body districts and disease skin states (i.e., atopic dermatitis, psoriasis, acne). In fact, relatively little information is available about the composition of the healthy skin microbiota. The cosmetic industry is especially interested in developing products that maintain and/or improve a healthy skin microbiota. Therefore, in the present work, the authors chose to investigate in detail the structure and composition of the basal bacterial community of the face. Ninety-six cheek samples (48 women and 48 men) were collected in the same season and the same location in central northern Italy. Bacterial DNA was extracted, the 16S rDNA gene was amplified by PCR, the obtained amplicons were subjected to next generation sequencing. The principal members of the community were identified at the genus level, and statistical analyses showed significant variations between the two sexes. This study identified abundant members of the facial skin microbiota that were rarely reported before in the literature and demonstrated the differences between male and female microbiota in terms of both community structure and composition.

## 1. Introduction

A healthy microbiota is traditionally associated with a healthy ecosystem [1,2,3,4], and this relationship prompted numerous studies on the human microbiota and its role in health and disease [5,6,7]. This collection of microorganisms contributes actively to the immune system efficacy, and microbiota dysbiosis has been linked to various diseases [3,8]. However, most studies to date focused either on whole-body characterization or on common skin diseases (atopic dermatitis, psoriasis, acne) [9,10,11,12], and it can be challenging to find detailed studies rendering the composition of a normal cutaneous microbiota or the indicators for altering conditions [13]. In fact, while the gut and skin ecosystems have been vastly studied [14,15], the high levels of variations between individuals makes difficult the identification of a standard, healthy microbiota.

This is particularly true for the skin: this habitat of 1.8 m^2^ is composed of a large variety of niches, each hosting a different microbial population [16]. It is also commonly accepted that both genetics and environmental factors are key players in shaping and maintaining the microbiota [9,13,15,17,18]. In fact, the high interpersonal variability is partly due to genetics, but also to a myriad of other factors that can be both intrinsic (sex, age, hormone levels) and extrinsic (diet, weather, environment) [19]. On one hand, whole-body studies demonstrated that skin appendages (sweat glands, hair follicles, sebaceous glands, etc.) and their secretions actively contribute to the skin’s ecosystem [10,15,20]. The density of such elements varies depending on the anatomic location, thus creating different micro-environments (humidity and salt levels, pH, nutrient availability, etc.) that modulate the structure and composition of the microbiota [21,22]. On the other hand, the skin has a different exposome depending on the geographical niche, and zones often in contact with the external environment (hands, face) are even more subject to variations than more sheltered areas (armpit, navel) [23]. Finally, it was demonstrated that the same locations are more similar between different subjects than to other locations on the same person [16]. Therefore, studies evaluating the influence of intrinsic and extrinsic factors on the skin microbiota should be carefully designed and focus on a specific body district.

Early studies on the microbiota’s composition tended to underestimate its diversity due to the limitations of the then-available investigative methods. Until the 2000s, most studies were conducted following culture-based methods that failed to accurately represent the microorganisms, mostly because of the impossibility to reproduce the exact microenvironment that allows the proliferation of organisms with multiple different preferred growth conditions [24,25,26]. However, the advent of Next Generation Sequencing (NGS) technologies in the past decades has allowed to overcome these limitations: by sequencing the genome of the microbiota (or microbiome) in a given sample, the sequencing of representative DNA sequences allowed to identify a wider variety of microorganisms [6,10,12]. Whole genome sequencing techniques, for example, identify all the live bacteria, fungi, viruses, and archaea of the microbiota, but analyses are quite costly, especially when handling a large number of microbiota samples [27,28]. DNA regions analyses are also permitted by cheaper alternatives such as pyrosequencing or 16S rDNA gene sequencing but allow to identify only the bacteria within a sample and does not differentiate between live and dead organisms [29,30]. Nevertheless, the metabarcoding of the 16S rDNA approach is considered very reliable to study the bacterial members of the microbiome and is vastly used in the scientific community [9,27]. Since bacteria compose most of the microflora, its accurate resolution to the genus level (and, to a lesser extent, the species level) allows to study even minor population variations and the ways they can be influenced by various extrinsic and intrinsic parameters [31,32].

Whole-body studies have demonstrated that the skin microbiota is composed of four main phyla (*Actinobacteria*, *Firmicutes*, *Proteobacteria*, and *Bacteroidetes*) [33] and is particularly dominated by the *Cutibacterium* (previously *Propionibacterium)*, *Corynebacterium*, and *Staphylococcus* genera [34]. However, the cutaneous microbiota is not constant across all body sites and its composition is highly dependent on the geographical niche microenvironment: while dry areas principally host *Proteobacteria*, moist sites can be dominated either by *Corynebacteria* spp., *Staphylococci* spp., or *Proteobacteria*, and *Cutibacteria* spp. are predominant on sebaceous sites such as the face [9,10,16]. There, the *Cutibacterium* genus generally represents about half of the total population and the *Corynebacterium* and *Staphylococcus* genera amount to 20–40%, while the other identified resident members usually belong to the *Micrococcus* (*Actinobacteria*), *Lactobacillus* (*Firmicutes*), and *Enhydrobacter* (*Proteobacteria*) [33,34]. Other genera such as *Streptococcus* and *Anaerococcus* (*Firmicutes*), *Pseudomonas* and *Sphingomonas* (*Proteobacteria*), and *Microbacterium* (*Actinobacteria*) have also been detected more recently [18,19,35]. However, the important influence of endo- and exogenous factors added to the high interpersonal variability does not allow to establish the general composition of the basal microbiota of the face [19,35,36]. Additionally, most studies focusing on the facial cutaneous microbiota have been conducted in eastern Asia and information is scanty for the western populations [37].

In an effort to accurately describe the standard cutaneous microbiota of the face in occidental subjects, in this study, we propose the detailed analysis of the 10 most abundant genera detected in our samples and the ways in which the volunteer’s sex influences their relative abundance. We decided to focus our analysis on the cheek of our subjects because of the particularly important differences in skin appendage density in this area and our data is in agreement with the literature [16,38,39,40,41,42,43,44].

## 2. Materials and Methods

### 2.1. Study Design

From a pool of over 300 samples from both male and female volunteers, we matched one-to-one 96 samples from the cheek (48 women and 48 men) based on season and living area to limit the influence of these factors. The tested subjects were aged 25–71 years (A = 25–35, B = 36–45, C = 46–55, D = 56–65, E = 66–75 years old), all Caucasian and residing in central northern Italy, and presenting no dermatological conditions in the sampled area. Smoking status was not considered, but most of our volunteers do not smoke (about 85%). The subjects under antibiotic therapy were excluded from the study. All participants enrolling in studies for Complife provide their informed consent upon participating to said studies. Samples were collected by trained technicians from Complife Italia Spa (Garbagnate Milanese, Italy).

### 2.2. Sample Collection

The participants were asked to refrain from washing their face 8–12 h prior to sampling, and to gently rinse and dry it upon collection. Samples were collected by swabbing an area defined by a 3.5 by 5.5 cm adhesive paper template (Copan Spa, Brescia, Italy). The FLOQswab^®^ (Copan Spa, Brescia, Italy) was humidified on one side with a drop of water and rubbed ten times horizontally inside the collection area. The swab was turned 180° and the dry side rubbed ten times vertically inside the template, then placed in 2 mL of preservation medium (eNAt^®^, Copan Spa, Brescia, Italy) and stored for up to four weeks at room temperature. Upon arrival in the genomics laboratory, the head of swab was placed inside a NAO basket^®^ (Copan Spa, Brescia, Italy), and successive eNAt^®^ addition, vortexing, and centrifugation steps resulted in the raw microbiota extract, stored at −20 °C until further use. Then, 500–700 µL of eNAt^®^ were added to the basket containing the swab head, and the 2 mL tube + basket device was centrifuged 1 min at 10,000× *g*. When the total volume of eNAt^®^ to be transferred was >1 mL, the flow-through was transferred in a clean 2 mL tube, and the process above repeated until no eNAt^®^ would remain in the collection tube. All liquid handling was performed under a laminar flow hood.

### 2.3. DNA Processing

Genomic DNA was extracted and purified using the QIAamp^®^ DNA Microbiome Kit (Qiagen, Hilden, Germany) from 0.5–1 mL of conservation medium following the provider’s guidelines. The bacterial 16S rDNA library was prepared by amplifying the V1-V3 hypervariable regions and indexing each sample for further identification (Microbiota Solution A, Arrow Diagnostics, Genoa, Italy). Quality control of the samples was conducted through (i) DNA quantification after extraction and during library preparation (Qubit^TM^ Flex Fluorometer, Thermo Fisher Scientific, Waltham, MA, USA) and (ii) electrophoresis in 1.5% agarose gel in TAE 1X after each PCR round for control of the amplicon size and purity. Blank samples were tested when establishing the general extraction procedure. A negative control was included to each PCR, and the absence of contamination was controlled via electrophoresis gel (along with the correct amplification of the bacterial DNA).

Gene sequencing was conducted on the MiSeq platform using the MiSeq^®^ v2- or Nano v2-500 cycle Reagent Kits and PhiX as an internal standard, both supplied by Illumina Inc. (San Diego, CA, USA), as also reported in Torre and coworkers [45].

### 2.4. Microbiome and Statistical Analyses

Raw sequences were processed by MicrobAT (Microbiota Analysis Tool) v. 1.1.0 software (SmartSeq Srl, Novara, Italy). The software specifies the Phylum, Class, Order, Family, Genus, and Species of the bacteria found in the samples. As previously reported in Bona et al. 2021 [46] and Torre et al., 2022 [45], MicrobAT is based on the RDP database (v.11.4) and it does not produce OTUs (operational taxonomic units). In particular, obtained sequences, after being filtered for length and quality (data quality evaluation), were aligned against the RDP database, and were assigned to a specific species if they meet the following criteria: query coverage 80% and similarity 97%. Moreover, from MicrobAT, three files were generated and processed by Microbiome Analyst software. First, data filtering was used in order to identify and remove features that are unlikely to be useful when modeling the data. Features having low count and variance can be removed during the filtration step while those having very few counts are filtered based on their median abundance levels (minimum count 4) across samples (prevalence).

The biodiversity within and between our groups was calculated using the phyloseq package [40]: the alpha diversity was characterized via the richness by measuring the Shannon index (*H’*, total number of taxa), and the evenness with the Simpson index (distribution of abundances) [47]. The beta diversity analysis method was used to compare the microbial community composition of each group, generating a distance (or dissimilarity) matrix. Measurements were performed using Bray–Curtis dissimilarity, while graphical representation of the matrices was obtained using Principal Coordinate Analysis (PCoA). The statistical significance of the clustering model of the sorting graphs was assessed using permutational analysis of variance (PERMANOVA). Statistical differences in taxa abundance between the male and female groups were assessed by applying the Linear Discriminant Analysis (LDA) Effect Size (LefSe) method [41], with a default *p*-value cut-off of 0.05.

Principal component analysis was performed using all the considered the most abundant phyla and species (relative abundance > 0.2%). Finally, two-way ANOVA was used to discriminate the effects of the two factors (“sex” and “age”) and of their interaction (“sex × age”). Differences were considered significant for *p*-values < 0.05. These analyses were performed using R (v. 3.5.1) R Core Team [48] using FactoMineR and Factoextra packages.

## 3. Results

### 3.1. Community Profiling

Because sequences in 16S rDNA metagenomic studies are generally too short to achieve accurate species affiliation [49], the analysis was limited to the genus level. The observed microbiota consisted of three different phyla, Firmicutes, Proteobacteria, and Actinobacteria.

Multivariate analysis, performed at phylum level, showed that the skin microbiota in males and females was different and strongly modulated by two phyla: Firmicutes influenced mainly the male skin microbiota, while Proteobacteria influenced the female one (Figure 1A). On the contrary, when labeled by age category or considering the combination of gender and age, the samples resulted to be less separated according to dimensions 1 and 2. These observations are also confirmed by the two-way ANOVA (Figure 1D) which showed that the sex factor had a significant influence both in Firmicutes and Proteobacteria, while the age factor was never significant.

Multivariate analysis performed at genus level is presented in the Figure 2. As previously observed at phylum level, the analysis at genus level confirms that the microbiota is mostly affected by sex rather than by age factor. This observation is also confirmed by the two-way ANOVA; results reported in Table 1. Considering this preliminary statistical analysis results that clearly identified sex as determinant factor in the variability explanation of the microbiota composition in our cohort, the following analyses will be presented only for this factor.

#### 3.1.1. Alpha-Diversity

Richness was assessed using the Shannon index (*H’* = ln(total number of taxa, *H’* ≥ 0), and evenness using the Simpson index (distribution of abundances, 0 ≤ D ≤ 1). Because of the high interpersonal variability, the data might not follow a normal distribution, so the non-parametric tests, Kruskal–Wallis and Mann–Whitney, were preferred [50]. The diversity distribution in the two survey groups was graphically represented as boxplots (Figure 3), the values of interest summarized in Table 2, and the populations were considered statistically different for *p*-values < 0.05. The diversities calculated as functions of the Shannon (Figure 3a and Table 2) and Simpson (Figure 3b and Table 2) indexes showed statistically significant differences between men and women. 

Overall, the Shannon index absolute values are quite low (particularly at the phylum level; Figure 3a and Table 2), translating that only a few members will represent most of the microbiota. At the phylum level, men and women have neighboring minimum and maximum values with a comparable spread and similar dispersion and distribution (Table 2), indicating a similar range of taxa represented. However, *p* < 0.05 demonstrates fundamental membership differences between the two groups. At the genus level, the lower *p*-value further emphasizes the statistical differences between the two sexes. The lower median in men, accompanied by lower minimum and maximum values, highlight the overall higher biodiversity in women. Finally, despite both having similar dispersions (see spread), the higher distribution in women indicates higher microbiota interpersonal variability in this group.

In the analysis of the Simpson index, the *p*-values are considerably lower than 0.05 at any taxonomic level (data not shown), which further confirms the radical structure differences between male and female facial microbiotas. Similar to the *H’* results, the female microbiome appears more diverse (higher median, Figure 3b and Table 2) and distributed (longer box, Figure 3b) than the male one, despite both having similar dispersions (spread, Table 2). With a right-skewed, closer to 1 (at the genus level) median diversity, it appears that the taxa found in the female microbiota are more diverse from one volunteer to another compared to the male microbiota.

#### 3.1.2. Beta-Diversity

The PCoA plot (Figure 4) shows the different distribution of men and women skin microbiotas within our cohort, with the ellipses indicating confidence intervals of 95%. The partly overlapping (but not superimposed) ellipses and the low *p*-values (<0.001 independently of the taxonomic level, data not shown) indicate a contrasting microbiota composition between the two sexes, which confirms the structural differences already suggested by the α-diversity (Figure 3) results and multivariate analysis (Figure 1 and Figure 2). 

### 3.2. Community Composition

#### 3.2.1. Taxa Abundance

The MicrobiomeAnalyst tool detected 3 phyla, 6 classes, 13 orders, 20 families, 22 genera, and 71 species.

On one hand, the high read numbers of *Firmicutes* in men and of *Proteobacteria* in women is substantiated by their relative percentages, and on the other hand, *Actinobacteria* is present in similar proportion in both sexes, despite the higher read numbers in male subjects (Table 3).

The focus was put on the 10 most abundant genera which together represent more than 80% of the bacterial members of the microbiota (Table 4 and Figure 5). In the interest of clarity, the close parent taxa with % < 0.45 for both groups were removed from the graphical representation (proportion of *Acetobacter*, 10th most abundant genera). The taxa of interest were distributed into three genera belonging to *Actinobacteria* (*Propionibacterium*, *Corynebacterium*, and *Microbacterium*); four to *Proteobacteria*—two *Alphaproteobacteria* (*Sphingomonas* and *Acetobacter*), one *Betaproteobacteria* (*Pelomonas*); and one *Gammaproteobacteria* (*Pseudomonas*); three to *Firmicutes*—*Staphylococcus* (order), *Streptococcus* (*Lactobacillales* order), and *Anaerococcus* (*Clostridia* class). 

#### 3.2.2. LefSe Analysis of Signature Taxa

The LefSe analysis was performed at the phylum level to identify the statistically different taxa between our groups (|LDA score| > 2.0, *p*-value < 0.05; Table 3 and Figure 6) and to estimate the effect size of each significantly different taxon [40]. The high *p*-value for *Actinobacteria* (*p* = 0.247, Table 3) compared with the similar relative abundances in the two groups confirms the removal of bias linked to read numbers. Similarly, the *p*-values are well below 0.05 for both *Proteobacteria* (*p* = 7.40E-09) and *Firmicutes* (*p* = 4.62E-05), validating the differences in community composition between the two study groups.

The same statistical analysis was performed at each lower taxonomic level (class, order, family; data not shown) down to the genus level, and the results overall followed the same trend. The only exception appeared from the order level: *Lactobacillales* members (and, more specifically, the *Streptococcaceae* family) are expressed significantly more in females than in males. A heat tree analysis was also performed (data not shown) and confirmed the results of the LefSe analysis.

The LefSe analysis performed at the genus level (Table 5 and Figure 7) revealed that 15 out of the total 22 identified genera were significantly different from one sex to the other (|LDA score| > 2.0, *p*-value < 0.05). Among them, eight out of the ten most abundant genera showed sex-specific variations (Table 5, in bold), and the members of the *Proteobacteria* family were often accompanied by other entities from the same family (*Blastomonas* and *Sphingomonas*, *Aquabacterium* and *Pelomonas*, and *Enhydrobacter* and *Pseudomonas*). Similarly, four of the highlighted genera were accompanied by at least one other member of the same order: *Ralstonia* and *Pelomonas* (and *Aquabacterium*) belong to the *Burkholderiales* order, and *Propionibacterium*, *Corynebacterium*, and *Microbacterium* to the *Actinomycetales*. Interestingly, two *Micrococcaceae* were also identified in the *Actinomycetales* order (*Arthrobacter* and *Micrococcus*), confirming the relevancy of the family’s presence in data analysis.

## 4. Discussion

In the past decades, the introduction of novel DNA sequencing and analysis systems in genomics and metagenomics has allowed to take a close look at the microbiota and its role in human health and disease [19,27]. These technology advances particularly benefitted the field of skin microbiota studies since it allowed to overcome the difficulties traditionally met with culture-based approaches [10,16,51]. Today, researchers have access to plethora of data that can be challenging to interpret, and we are just now scratching the surface of what constitutes a healthy microbiota: in addition to the complex interactions between the cutaneous matrix and its inhabitants, the microbiota’s population is remarkably diverse and its structure is highly subject to inter- and intra-personal variations [13,18].

The goal of this study was to pinpoint the pattern differences between the male and female facial microbiomes and our team decided to focus on the cheek. First, because this area presents distinct skin appendage densities between the two sexes: in fact, our data is in agreement with the literature [16,38,39], and we observed that the differences in community structure between males and females reflect the physiological and anatomical distinctions between the two sexes [40,52]. Second, the face is constantly exposed to its surroundings, thus studying this area will allow further investigations into the effect of environmental factors on the cutaneous microbiota [18,40]. In the future, these results should also allow to better observe if men’s and women’s microbiomes behave similarly under comparable circumstances.

Alpha-diversity was first evaluated to determine the number of members present in the face microbiota (taxa richness) and the ways in which they are distributed (taxa evenness). Independently of the chosen indexes and in agreement with previous studies, men consistently displayed a reduced alpha-diversity at each taxonomic level [39,53,54]. Biodiversity being dependent on nutrient availability and variety, the traditionally higher sebum levels in men and the consequent anoxic environment naturally exclude the presence of aerobic members while reinforcing the monopoly of anaerobic bacteria such as *Cutibacterium* spp. [34,55]. The low numbers could also be explained by the acidification of the skin surface induced by the release of (i) free fatty acids (FFA) from the sebaceous glands, and (ii) lactic acid from sweat glands [41,55], both present in higher numbers in men. The higher biodiversity in women should, in turn, be attributed to their thinner skin, more acidic pH and reduced sweat production [39], but could also be a consequence of better skin hydration and more substantial use of cosmetic products [16,55,56,57,58]. In fact, the recurrent washing and regular chemical and mechanical exfoliation of the skin accelerate the renewal of the skin layers, thus continuously changing the population of transient species—which would also explain the higher interpersonal variability in women compared to men.

If the alpha-diversity values indicate that women present a greater bacterial diversity while the male’s microbiota seem dominated by a few members, the beta-diversity results imply that each sex hosts different members of the bacterial community: the relative abundances of *Actinobacteria* are similar between both groups, but the second most abundant phylum in males is *Firmicutes* and *Proteobacteria* in females. The statistical relevance of the results was demonstrated by LefSe analysis, and while the presence of only these three phyla is in line with published results [10,27], not all researchers agree on whether men and women host similar bacterial community. In fact, while some did not observe sex-specific fluctuations on the inner wrist [59], the nares [60], or the forehead and whole body [61], other studies noted differences through study of the hands [53] or even of the whole body [39].

A heat tree analysis was also performed at the genus level to confirm the results obtained with the LefSe analysis (data not shown). The same taxa were highlighted and even classified in the same order with analogous *p*-values, with *Staphylococcus* and *Anaerococcus* majoritarian in men and the others preponderant in women. The graphical representation (data not shown) also emphasized that the differences observed in females and males were driven by a greater bacterial diversity in the female group.

As expected from the low *H’* values, observation of the taxa abundance at the genus level reveals that only a few members of the flora make up for the majority of the microbiome: the four genera *Cutibacterium* and *Corynebacterium* (*Actinobacteria*), *Sphingomonas* (*Proteobacteria*), and *Staphylococcus* (*Firmicutes*) alone account for more than 75% in both sexes. As typically documented in human cutaneous microbiota studies, *Cutibacterium* spp. account for about half of all microbial flora for both sexes [16], while *Staphylococcus* spp. are quite prominent [62] but do not hold the same importance in men and women. In fact, the second most abundant genus in female volunteers is *Sphingomonas*, an *Alphaproteobacteria* which has rarely been described as a common member of the basal microbiome of the skin [19]. 

The high proportion of *Cutibacteria* spp. (and *Actinobacteria* members in general) is explained by their preference towards sebaceous areas such as the face, chest and back [16], but the LefSe analysis did not reveal relevant differences between the two groups for its principal members *Cutibacterium* and *Corynebacterium*. Variations in sebum levels, on the other hand, might affect other aerobic or facultative anaerobic bacteria (*Staphylococci*, *Sphingomonas*, *Pseudomonas* and *Microbacteria* spp.). The *Anaerococcus* genus, for example, is reportedly found in the human microbiota and associated to diseased states [63], but its superior abundance in men should be related to its anaerobic properties [64] rather than to a reduced hygiene. However, in the case of *Staphylococcus*, its higher fraction in male microbiota should preferably be attributed to the greater sweat secretion in men [65]. Similarly, the larger proportion of *Microbacterium* in women could be explained by the salt levels since its growth can be inhibited by high concentrations [66], but could also be attributed to the presence of heavy metals [67] in make-up products, subject of growing concern in the cosmetic industry. The only *Firmicute* present in bigger proportion in women than in men was the *Streptococcus* genera: although *Lactobacillus* is more commonly expected (unpublished results, [34,68]), our results are in line with other reports [34,61] suggesting that members of the *Lactobacillales* order adopt the same behavior independently of their genera.

Members of the *Proteobacteria* phylum are more commonly accepted as environmental bacteria, which tends to confirm that the frequent depletion and repopulation of the microbiota in women leads to exposome-driven bacteria recolonization. In fact, *Sphingomonas* (along with *Acetobacter* and most members of the *Alphaproteobacteria* class) is long associated with plant colonization [69] and was reported only recently as a cutaneous microbiota member [56]; its relation to the microbiome’s balance and the presence of pollutants should be investigated [70]. To our knowledge, the genus *Pelomonas* was reported only recently as well, and appears to be found mostly in the dermal compartment (rather than the skin surface in our case) where it is considered a core commensal [71]. *Pseudomonas*, on the other hand, is often reduced to its pathogenic members and is a part of the normal adult gut microbiota [72], but its presence as a normal member of the facial flora was described in the latest works of Kim et al. in 2021 [40]. Similarly to both *Pelomonas* and *Anaerococcus*, its presence is expected in deeper compartment of the skin [71,73], but our results suggest they are also capable to survive in the more superficial layers as commensal members.

If *Corynebacteria* and *Micrococci* spp. are commonly accepted as regular minor members of the microbiota [19], our results demonstrate that other low-abundance members should be investigated as well. In fact, our research suggests that *Lactobacilli* spp. are substituted by *Streptococci* spp. in the facial microbiota, and that *Sphingomonas*, *Pelomonas*, *Pseudomonas*, and *Anaerococci* spp. warrant further investigations, as well as members of the *Neisseriaceae* family. Finally, LefSe analysis at the family level (data not shown) demonstrated that both the *Micrococcaceae* and *Neisseriaceae* families are statistically more expressed in the female microbiome, but no descendants at the genus level could be identified. This might be attributed to the low resolution traditionally associated to 16S rRNA metagenomic studies but could also be a consequence of lacking reference genomes to be compared to.

## 5. Conclusions

We observed that the differences in community structure between males and females reflect the physiological and anatomical distinctions between the two sexes [52] through specific variations of the commonly described *Cutibacterium*, *Staphylococcus*, *Corynebacterium*, *Microbacterium*, and *Streptococcus* genera [9,16,36]. We also confirmed the presence as resident taxa of rarely described members belonging to the phyla *Proteobacteria* (*Sphingomonas* and *Acetobacter*, *Pelomonas*, *Pseudomonas* genera) and *Firmicutes* (genus *Anaerococcus* from the *Clostridia* class) [19,34]. These observations now provide us with a clear and well-defined framework for future studies: we successfully demonstrated that sex is a crucial factor in shaping the microbial community of the human face, and the identification of low-abundance, high-relevance entities suggests that factor-driven microbiota variations do not necessarily depend on the most abundant members of the microbial flora. Continuous sample collection will also guide us into evaluating the influence of other parameters on the cutaneous bacterial population and to refine the definition of the normal microbiome of the human face.

In fact, unpublished preliminary results indicate that several endogenous and exogenous factors induce significant changes in either sex. In the future, the differences observed between male and female subjects could become the cornerstone for developing more personalized skincare, and observation of other variables (age, hormone levels, etc.) will allow to gain a better understanding of the interdependent relationship between a given microbiota and its host. Finally, the constant exposure of the face to the external environment should allow to study the influence of exogenous parameters such as seasonality or atmospheric pollutants.

## Figures and Tables

**Figure 1 microorganisms-10-02470-f001:**
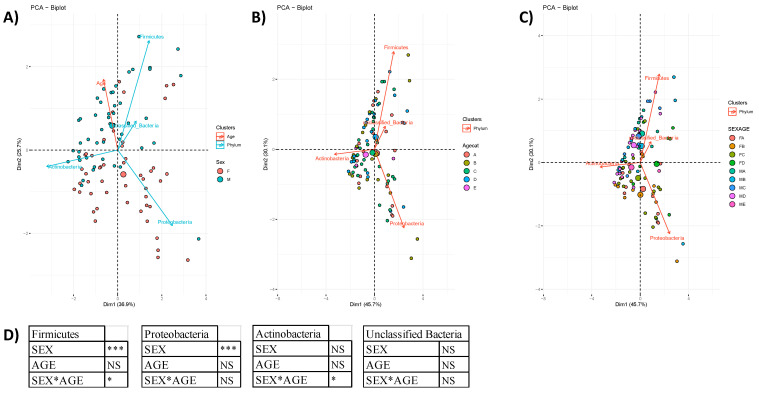
PCA analysis at Phylum level based on the following: (**A**) Sex (62.6% of variability explained by dimensions 1 and 2); (**B**) Age categories (A: 25–35 years old, B: 36–45, C: 46–55, D: 56–65, E: 66–70; 75.8% of variability explained by dimensions 1 and 2); (**C**) Sex + Age categories, 75.8% of variability explained by dimensions 1 and 2). The largest ball represents the mean of the samples, while the small dot indicates the single sample. (**D**) Two-way ANOVA results for the phyla Firmicutes, Proteobacteria, Actinobacteria and Unclassified bacteria considering the two factors “Sex”, “Age” and their interaction “Sex*Age”. NS = Not Significant; * = Significant at *p* < 0.05; ** = Significant at *p* < 0.01; *** = Significant at *p* < 0.001.

**Figure 2 microorganisms-10-02470-f002:**
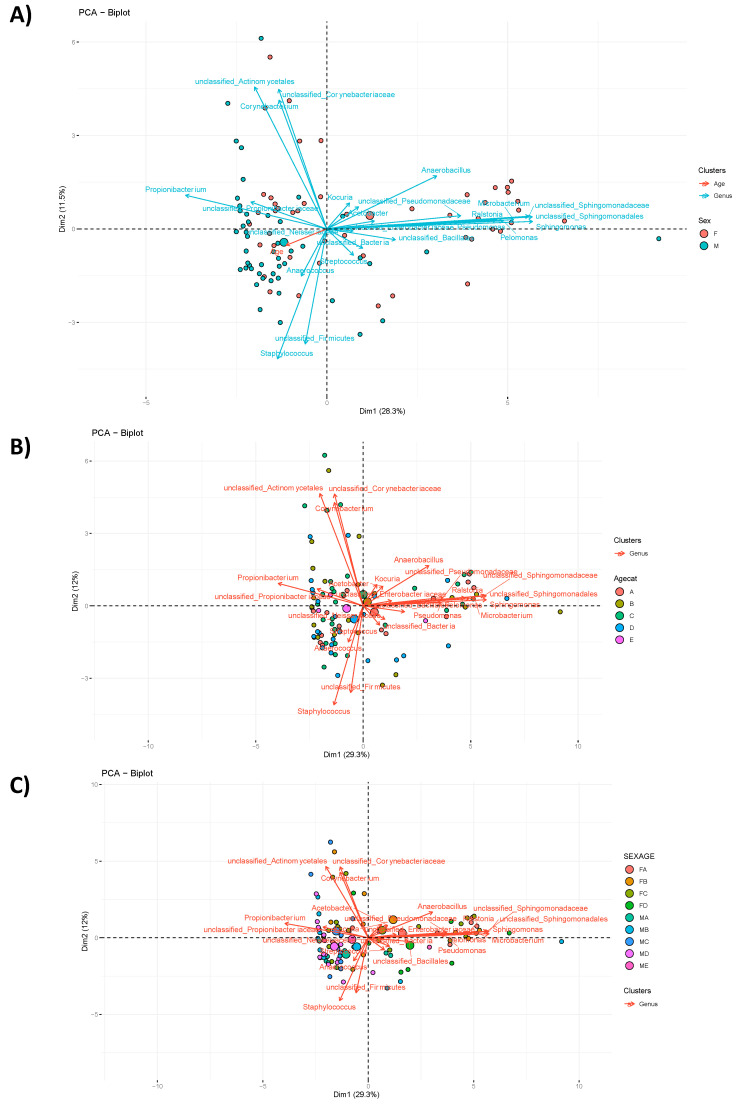
PCA analysis at Genus level based on the following: (**A**) Sex (39.8% of variability explained by dimensions 1 and 2); (**B**) Age categories (A: 25–35 years old, B: 36–45, C: 46–55, D: 56–65, E: 66–70; 41.3% of variability explained by dimensions 1 and 2); (**C**) Sex + Age categories, 41.3% of variability explained by dimensions 1 and 2. The largest ball represents the mean of the samples, while the small dot indicates the single sample.

**Figure 3 microorganisms-10-02470-f003:**
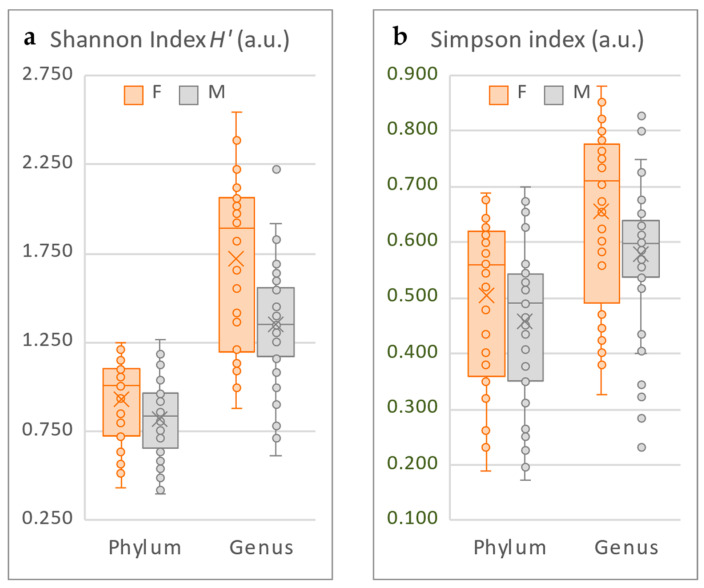
Graphic representation of the α-diversity at the phylum and genus taxonomic levels, calculated using the Shannon (**a**) and Simpson (**b**) indices. a.u. = arbitrary unit.

**Figure 4 microorganisms-10-02470-f004:**
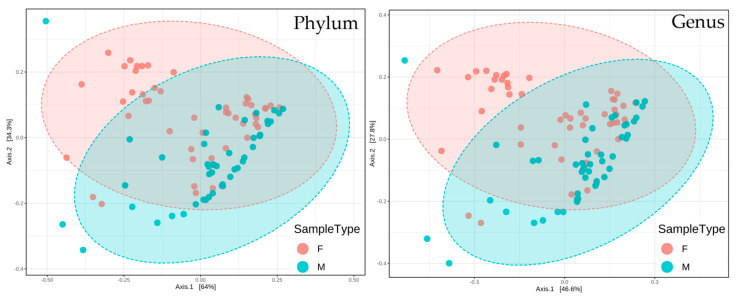
Graphic representation of β-diversity at the phylum and genus taxonomic levels.

**Figure 5 microorganisms-10-02470-f005:**
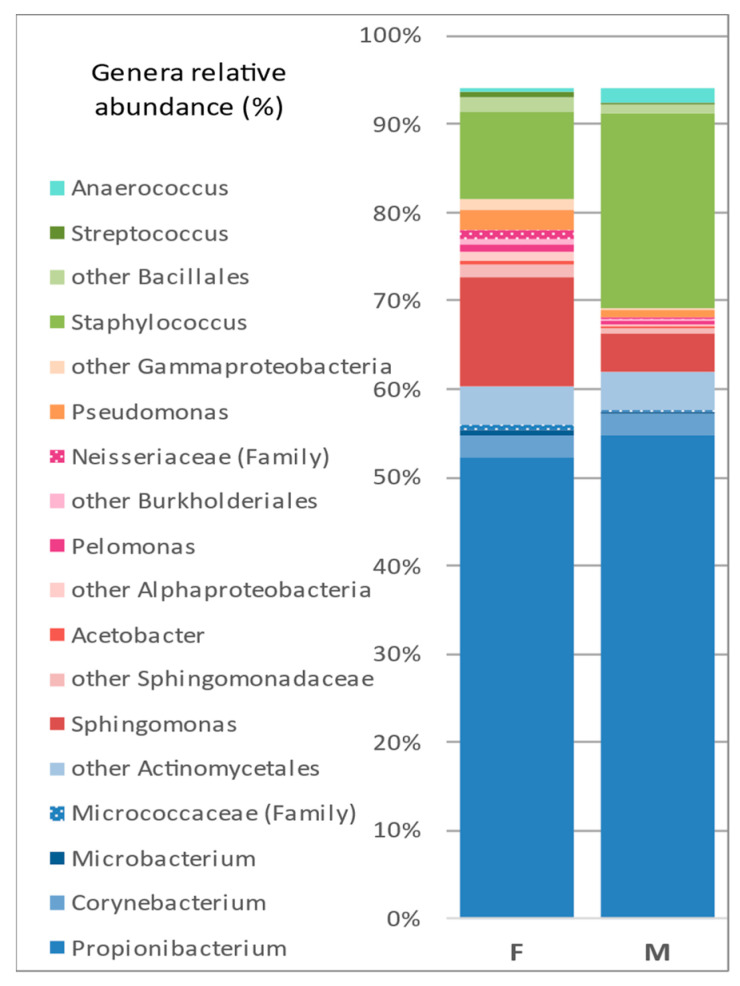
Bar graph of the most abundant genera and their most representative parent taxa.

**Figure 6 microorganisms-10-02470-f006:**
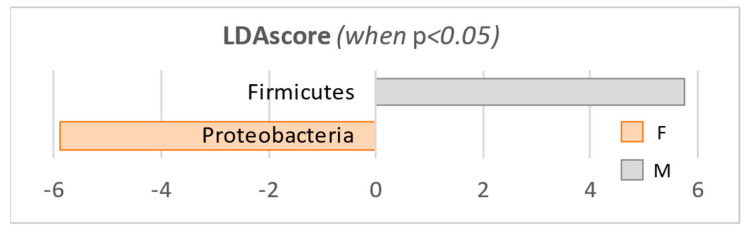
Results obtained by the LefSe analysis, at phylum level.

**Figure 7 microorganisms-10-02470-f007:**
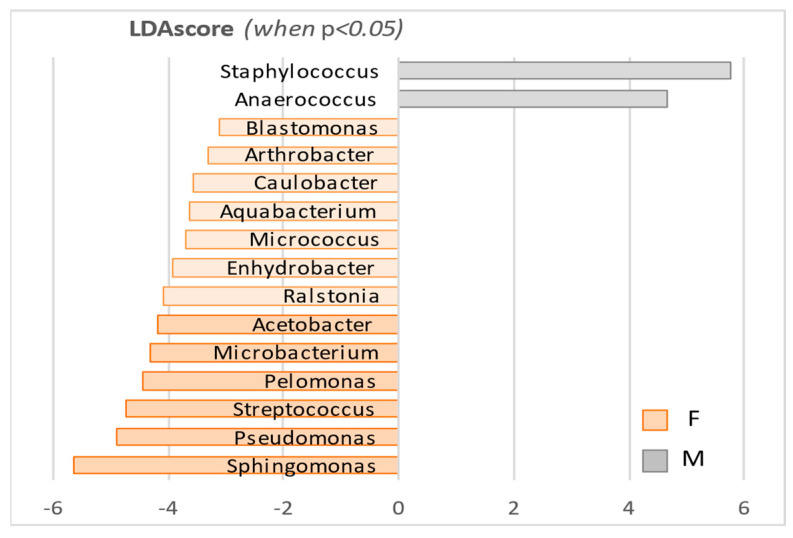
Graphical representation of results obtained by the LefSe analysis, at genus level.

**Table 1 microorganisms-10-02470-t001:** Two-way ANOVA results for the most represented genera considering the two factors “Sex”, “Age” and their interaction “Sex*Age”. NS = Not Significant; * = Significant at *p* < 0.05; ** = Significant at *p* < 0.01; *** = Significant at *p* < 0.001.

	Sex	Age	Sex*Age
Acetobacter	**	NS	*
Anaerobacillus	*	NS	NS
Anaerococcus	**	NS	NS
Corynebacterium	NS	NS	NS
Microbacterium	***	NS	NS
Pelomonas	**	NS	NS
Propionibacterium	NS	NS	**
Pseudomonas	***	**	**
Ralstonia	**	NS	NS
Sphingomonas	***	NS	NS
Staphylococcus	***	NS	*
Streptococcus	NS	NS	NS
unclassified_Actinomycetales	NS	NS	NS
unclassified_Bacillales	**	NS	NS
unclassified_Bacteria	NS	NS	NS
unclassified_Corynebacteriaceae	NS	NS	NS
unclassified_Firmicutes	**	NS	*
unclassified_Neisseriaceae	*	NS	NS
unclassified_Propionibacteriaceae	NS	NS	NS
unclassified_Sphingomonadaceae	***	NS	NS
unclassified_Sphingomonadales	***	NS	NS

**Table 2 microorganisms-10-02470-t002:** Values of interest at the phylum and genus levels, calculated from Shannon and Simpson indexes.

		*p*-Value	MW/KW ^1^	Median		Spread	
				F	M	F	M
**Shannon**	*Phylum*	7.67E-03	1514	1.009	0.839	0.573	0.442
	*Genus*	2.28E-04	1647	1.888	1.349	1.008	0.731
**Simpson**	*Phylum*	2.29E-02	1462	0.560	0.491	0.501	0.528
	*Genus*	3.63E-03	1546	0.709	0.597	0.553	0.595

^1^ Mann–Whitney/Kruskal–Wallis non-parametric test.

**Table 3 microorganisms-10-02470-t003:** Results of taxa abundance and LefSe analyses.

		Number of Reads	Phyla Relative Abundance	*p*-Values	FDR	LDA Score
Proteobacteria	**W**	2,152,800	21.9%	7.40E-09	2.96E-08	−5.89
	**M**	616,800	7.4%			
Firmicutes	**W**	1,514,800	13.2%	4.62E-05	9.24E-05	5.75
	**M**	2,638,200	25.7%			
Actinobacteria	**W**	5,878,800	60.3%	2.47E-01	3.29E-01	5.26
	**M**	6,246,500	62.0%			

**Table 4 microorganisms-10-02470-t004:** Relative abundance of the most abundant genera and of their most representative parent taxa.

Phylum	Class	Order	Family	Genus	F	M
Actinobacteria	Actinobacteria	Actinomycetales	Propionibacteriaceae	Propionibacterium	52.18%	54.65%
			Corynebacteriaceae	Corynebacterium	2.60%	2.61%
			Microbacteriaceae	Microbacterium	0.54%	0.16%
			Micrococcaceae (Family)		0.61%	0.23%
		other Actinomycetales			4.30%	4.25%
Proteobacteria	Alphaproteobacteria	Sphingomonadales	Sphingomonadaceae	Sphingomonas	12.48%	4.41%
			other Sphingomonadaceae		1.38%	0.50%
		Rhodospirillales	Acetobacteraceae	Acetobacter	0.45%	0.14%
	other Alphaproteobacteria				0.97%	0.33%
	Betaproteobacteria	Burkholderiales	Comamonadaceae	Pelomonas	0.88%	0.35%
		other Burkholderiales			0.62%	0.23%
		Neisseriales	Neisseriaceae (Family)		1.11%	0.33%
	other Betaproteobacteria				0.25%	0.06%
	Gammaproteobacteria	Pseudomonadales	Pseudomonadaceae	Pseudomonas	2.20%	0.71%
	other Gammaproteobacteria				1.24%	0.23%
Firmicutes	Bacilli	Bacillales	Staphylococcaceae	Staphylococcus	9.88%	22.12%
		other Bacillales			1.58%	1.00%
		Lactobacillales	Streptococcaceae	Streptococcus	0.77%	0.21%
		other Lactobacillales			0.19%	0.03%
	Clostridia	Clostridiales	Clostridia Incertae Sedis XI	Anaerococcus	0.40%	1.59%
	other Clostridia				0.08%	0.30%
uncl. Bacteria	uncl. Bacteria				4.60%	4.97%

**Table 5 microorganisms-10-02470-t005:** Results obtained by the LefSe analysis at the genus level. ^A^ Actinobacteria (phylum): * Micrococcaceae (family); ^α^ Alphaproteobacteria (class): ^•^ Rhodospirillales (order), ° Sphingomonadaceae (family); ^β^ Betaproteobacteria (class): ^‡^ Burkholderiales (order)^†^, Comamonadaceae (family); ^γ^ Gammaproteobacteria (class): ^p^ Pseudomonadales (order); ^F^ Firmicutes: ^B^ Bacillales (order), ^L^ Lactobacillales (order), ^C^ Clostridiales (order).

	*p*-Values	FDR	Median Read Number		LDA Score
			F	M	
**Microbacterium ^A^**	**3.63E-09**	**5.72E-08**	**55,004**	**13,319**	**−4.32**
Ralstonia ^β‡^	4.67E-09	5.72E-08	36,469	11,060	−4.10
Arthrobacter ^A^*	7.53E-09	7.38E-08	5514.1	1367.8	−3.32
**Sphingomonas ^α•^°**	**1.35E-08**	**9.34E-08**	**1,239,200**	**383,550**	**−5.63**
Blastomonas ^α•^°	2.41E-08	1.07E-07	3575.1	947.4	−3.12
**Pelomonas ^β‡†^**	**4.35E-08**	**1.78E-07**	**92,260**	**34,183**	**−4.46**
**Staphylococcus ^FB^**	**7.02E-07**	**2.29E-06**	**1,088,700**	**2,297,500**	**5.78**
Caulobacter ^α^	1.87E-05	4.16E-05	9776.9	2312.5	−3.57
Micrococcus ^A^*	5.88E-05	1.25E-04	13,784	3772.4	−3.70
**Pseudomonas ^γp^**	**8.32E-05**	**1.70E-04**	**207,110**	**43019**	**−4.91**
Aquabacterium ^β‡†^	1.04E-04	1.91E-04	14,808	6208.9	−3.63
Enhydrobacter **^γp^**	1.05E-04	1.91E-04	20,008	2465.2	−3.94
**Streptococcus ^FL^**	**2.03E-04**	**3.55E-04**	**132,430**	**23,639**	**−4.74**
**Acetobacter ^α^**	**2.91E-03**	**4.60E-03**	**48,747**	**17,091**	**−4.20**
**Anaerococcus ^FC^**	**3.54E-03**	**5.26E-03**	**52,957**	**146,280**	**4.67**
*Propionibacterium ^A^*	*1.68E-01*	*2.17E-01*	*5,004,700*	*5,385,600*	*5.28*
*Corynebacterium ^A^*	*5.73E-01*	*6.53E-01*	*301,120*	*349,850*	*4.39*

## Data Availability

Not applicable.

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
