# Peer review of "Influence of Sex on the Microbiota of the Human Face"

_microorganisms, 2022, doi:10.3390/microorganisms10122470_

Round 1

Reviewer 1 Report

Minor comments

- Abstract line 22: the samples were subject to amplicon sequencing after the 16S rrNA PCR, and not to genome sequencing- this is misleading

- Why no analyses were peformed on the sub-genus level? OTU, ASV

- Table 3: The percentage indicate median or mean values?

Major comments

Abstract line 16: "In fact, relatively little information is available about the composition of the healthy skin microbiota." Well this is no longer true, many studies described the skin microbiome in health, also in studies which investigated skin diseases such as atopic dermatitis, psoriasis and did recruit, and described the microbiome of healthy controls.

Abstract line 24: The authors claim to identify members of the facial skin microbiota that were not reported before in the literature; however this claim is not supported by the data presented, and authors do report in the discussion that identified taxa were described elsewhere.

Her is a study which examined skin microbiota from the cheek, and reports similar results:

Kim, HJ., Kim, J.J., Myeong, N.R. et al. Segregation of age-related skin microbiome characteristics by functionality. Sci Rep 9, 16748 (2019). https://doi.org/10.1038/s41598-019-53266-3

Lines 90 – 94: Studies on the facial cutaneous microbiota are lacking in western populations: A recent and a landmark study did investigate the face microbiome including from cheek samples, and should be acknowledged.

Hillebrand GG, Dimitriu P, Malik K, Park Y, Qu D, Mohn WW, Kong R. Temporal Variation of the Facial Skin Microbiome: A 2-Year Longitudinal Study in Healthy Adults. Plast Reconstr Surg. 2021 Jan 1;147(1S-2):50S-61S. doi: 10.1097/PRS.0000000000007621. PMID: 33347075.

Oh J, Byrd AL, Park M; NISC Comparative Sequencing Program, Kong HH, Segre JA. Temporal Stability of the Human Skin Microbiome. Cell. 2016 May 5;165(4):854-66. doi: 10.1016/j.cell.2016.04.008. PMID: 27153496; PMCID: PMC4860256.

Study design, lines 104-106: More detailed metadata need to reported. For example what was the age distribution across the groups. Did the authors take into consideration further recruitment criteria including use of any medication – especially antibiotics – smoking status? Additional criteria such as life style (sport, ...), diet, all of which could influence the skin microbiome.

Sample collection and DNA processing:

Blank samples: were negative collection controls (only buffers, sampling environment … etc.) included along with the skin samples during sampling? Same for DNA extraction negative controls? These controls are important for subsequent "decontamination" procedure

Library preparation: Positions and sequences of the primers, and PCR conditions need to reported. Were PCR negative controls (PCR reagents) checked? Dual or single primers barcoding was used? What was the final concentration of the pool? Were samples pooled to the same/equimolar concentration? How was the sequencing performed and the two sequencing kits used? The MiSeq® v2 kit is not suffiecnt to cover the entire V1-V3 length. Were positive controls (mock community of known composition) sequenced along?

Microbiome analyses:

Line 136: Which RDP classifier released version was used? Taxonomy may differ across versions

Lines 141-142:"Data was not filtered in order to retain singletons and doubletons from the occurrence table, which participate to a meaningful estimate" The state of the art approach – especially for low biomass organs- is to actually filter out not only singletons and doubletons, but also low abundant taxa, present at 1% or 2%. Numerous studies demonstrated that these low abundant taxa potentially generate misleading diversity profiles, an example study is the following:

Reitmeier, S., Hitch, T.C.A., Treichel, N. et al. Handling of spurious sequences affects the outcome of high-throughput 16S rRNA gene amplicon profiling. ISME COMMUN. 1, 31 (2021). https://doi.org/10.1038/s43705-021-00033-z

Lines 142-144: "diversity should be assessed on the raw data since samples are not compared to each other but evaluated individually or between groups" Samples were indeed compared to each other- that´s the principle of beta diversity measures. Normalizing sampling depth across samples, should at least be evaluated on the performed analyses. Please have a look at this study:

Weiss S, Xu ZZ, Peddada S, Amir A, Bittinger K, Gonzalez A, Lozupone C, Zaneveld JR, Vázquez-Baeza Y, Birmingham A, Hyde ER, Knight R. Normalization and microbial differential abundance strategies depend upon data characteristics. Microbiome. 2017 Mar 3;5(1):27. doi: 10.1186/s40168-017-0237-y. PMID: 28253908; PMCID: PMC5335496.

Removal of contaminants: The skin harboring a low microbial biomass, influence of contaminants is crucial, and need to be addressed. Several analytical tools are available which rely on the negative controls collected during samples collection, and data generation. Such procedure is described in this recently published skin microbiota study:

Barnes CJ, Clausen ML, Asplund M, Rasmussen L, Olesen CM, Yüsel YT, Andersen PS, Litman T, Hansen AJ, Agner T. Temporal and Spatial Variation of the Skin-Associated Bacteria from Healthy Participants and Atopic Dermatitis Patients. mSphere. 2022 Feb 23;7(1):e0091721. doi: 10.1128/msphere.00917-21. Epub 2022 Feb 23. PMID: 35196118; PMCID: PMC8865923.

Figure 2 shows a sub-group of females to be distinct from the remaining female and male donors, and appear to be driving the difference, and significance between the two groups. Did the authors investigate the underlying reasons: technical, batch effects ...etc. that could explain such a disparity within the female donors?

Lines 423 – 425: Which studies show that members of Proteobacteria are commonly considered to be environmental noise rather than true skin residents?

Lines 208-209: Authors claim that males harbor more bacteria as their read numbers are significantly higher (!). There is a confusion here between the bacterial biomass i.e. the total collection of bacterial material in a given sample, and the number of reads or sampling depth obtained through PCR and sequencing techniques. When pooled into a library for sequencing, samples need to be equimolar. If samples are sequenced with different starting concentrations, this will bias the subsequent analysis and thus comparison as some will be underrepresented and others underrepresented. In the sequencing run, sampling depth should be similar across samples.

To compare the total bacterial biomass across samples, techniques such cell sorting/counting, ddPCR with total bacteria primers on samples with identical concentrations can determine the bacterial biomass within a sample.

On the other hand, bacteria counts or relative abundances generated from high-throughput sequencing can be further corrected using the actual bacterial total biomass.

For further details, an example study:

Zemb O, Achard CS, Hamelin J, De Almeida ML, Gabinaud B, Cauquil L, Verschuren LMG, Godon JJ. Absolute quantitation of microbes using 16S rRNA gene metabarcoding: A rapid normalization of relative abundances by quantitative PCR targeting a 16S rRNA gene spike-in standard. Microbiologyopen. 2020 Mar;9(3):e977. doi: 10.1002/mbo3.977. Epub 2020 Jan 11. Erratum in: Microbiologyopen. 2022 Jun;11(3):e1305. PMID: 31927795; PMCID: PMC7066463.

Data not shown: Authors rely too often on undisclosed data to support claims. Data need to be reported, and discussed at least for review purposes.   

Author Response

Reviewer 1

Minor comments 

- Abstract line 22: the samples were subject to amplicon sequencing after the 16S rrNA PCR, and not to genome sequencing- this is misleading

The authors agree with the reviewer and modify the text.

- Why no analyses were performed on the sub-genus level? OTU, ASV

The bioinformatic analysis were performed at species level and obtained results were reported at genus level.

- Table 3: The percentage indicate median or mean values?

Table 3 showed relative abundance values in males and females calculated by microbiome analyst selecting in taxa resolution the option “merging small taxa with counts < 10 based on median”.

Major comments 

Abstract line 16: "In fact, relatively little information is available about the composition of the healthy skin microbiota." Well this is no longer true, many studies described the skin microbiome in health, also in studies which investigated skin diseases such as atopic dermatitis, psoriasis and did recruit, and described the microbiome of healthy controls.

The authors are in partial agreement with the reviewer. Indeed, it is true that in the works that evaluate the variations in the diseased microbiota, the controls are described, but there are no specific insights related to the factors that modulate the healthy microbiota.

Abstract line 24: The authors claim to identify members of the facial skin microbiota that were not reported before in the literature; however this claim is not supported by the data presented, and authors do report in the discussion that identified taxa were described elsewhere.

Her is a study which examined skin microbiota from the cheek, and reports similar results:

Kim, HJ., Kim, J.J., Myeong, N.R. et al. Segregation of age-related skin microbiome characteristics by functionality. Sci Rep 9, 16748 (2019). https://doi.org/10.1038/s41598-019-53266-3

The author thank the reviewer for the correction and suggestion and modify the text accordingly.

Lines 90 – 94: Studies on the facial cutaneous microbiota are lacking in western populations: A recent and a landmark study did investigate the face microbiome including from cheek samples, and should be acknowledged.

Hillebrand GG, Dimitriu P, Malik K, Park Y, Qu D, Mohn WW, Kong R. Temporal Variation of the Facial Skin Microbiome: A 2-Year Longitudinal Study in Healthy Adults. Plast Reconstr Surg. 2021 Jan 1;147(1S-2):50S-61S. doi: 10.1097/PRS.0000000000007621. PMID: 33347075.

Oh J, Byrd AL, Park M; NISC Comparative Sequencing Program, Kong HH, Segre JA. Temporal Stability of the Human Skin Microbiome. Cell. 2016 May 5;165(4):854-66. doi: 10.1016/j.cell.2016.04.008. PMID: 27153496; PMCID: PMC4860256.

Done

Study design, lines 104-106: More detailed metadata need to reported. For example what was the age distribution across the groups. Did the authors take into consideration further recruitment criteria including use of any medication – especially antibiotics – smoking status? Additional criteria such as life style (sport, ...), diet, all of which could influence the skin microbiome.

Smoking status was not considered, most of our volunteers don't smoke (about 85%).

The patients assuming antibiotic therapy were excluded from the study.

Sample collection and DNA processing:

Blank samples: were negative collection controls (only buffers, sampling environment … etc.) included along with the skin samples during sampling? Same for DNA extraction negative controls? These controls are important for subsequent "decontamination" procedure

Blank samples were tested when establishing the general extraction procedure, but not further: since we process a high number of samples, it would be too expensive and unpractical. A negative control is included to each PCR round, and the absence of contamination is controlled via electrophoresis gel (along with the correct amplification of the bacterial DNA).

Library preparation: Positions and sequences of the primers, and PCR conditions need to reported. Were PCR negative controls (PCR reagents) checked? Dual or single primers barcoding was used?

The method proposed by Arrow is a kit that did not provide this information.

What was the final concentration of the pool?

The final concentration of the pool was 5pM.

Were samples pooled to the same/equimolar concentration?

Yes, the samples were pooled at the same concentration.

How was the sequencing performed and the two sequencing kits used? The MiSeq® v2 kit is not suffiecnt to cover the entire V1-V3 length. Were positive controls (mock community of known composition) sequenced along?

The author add this information in the text.

Microbiome analyses: 

Line 136: Which RDP classifier released version was used? Taxonomy may differ across versions.

Done

Lines 141-142:"Data was not filtered in order to retain singletons and doubletons from the occurrence table, which participate to a meaningful estimate" The state of the art approach – especially for low biomass organs- is to actually filter out not only singletons and doubletons, but also low abundant taxa, present at 1% or 2%. Numerous studies demonstrated that these low abundant taxa potentially generate misleading diversity profiles, an example study is the following:

Reitmeier, S., Hitch, T.C.A., Treichel, N. et al. Handling of spurious sequences affects the outcome of high-throughput 16S rRNA gene amplicon profiling. ISME COMMUN. 1, 31 (2021). https://doi.org/10.1038/s43705-021-00033-z

We agree with the reviewer and we added some more details in the text.

Lines 142-144: "diversity should be assessed on the raw data since samples are not compared to each other but evaluated individually or between groups" Samples were indeed compared to each other- that´s the principle of beta diversity measures. Normalizing sampling depth across samples, should at least be evaluated on the performed analyses. Please have a look at this study:

Weiss S, Xu ZZ, Peddada S, Amir A, Bittinger K, Gonzalez A, Lozupone C, Zaneveld JR, Vázquez-Baeza Y, Birmingham A, Hyde ER, Knight R. Normalization and microbial differential abundance strategies depend upon data characteristics. Microbiome. 2017 Mar 3;5(1):27. doi: 10.1186/s40168-017-0237-y. PMID: 28253908; PMCID: PMC5335496.

OK

Removal of contaminants: The skin harboring a low microbial biomass, influence of contaminants is crucial, and need to be addressed. Several analytical tools are available which rely on the negative controls collected during samples collection, and data generation. Such procedure is described in this recently published skin microbiota study:

Barnes CJ, Clausen ML, Asplund M, Rasmussen L, Olesen CM, Yüsel YT, Andersen PS, Litman T, Hansen AJ, Agner T. Temporal and Spatial Variation of the Skin-Associated Bacteria from Healthy Participants and Atopic Dermatitis Patients. mSphere. 2022 Feb 23;7(1):e0091721. doi: 10.1128/msphere.00917-21. Epub 2022 Feb 23. PMID: 35196118; PMCID: PMC8865923.

Upon establishing our analysis pipeline, both the repeatability and reproducibility of the results were assessed. The results were compared against the literature to control the absence of pollutants possibly introduced during sample manipulation.

Figure 2 shows a sub-group of females to be distinct from the remaining female and male donors, and appear to be driving the difference, and significance between the two groups. Did the authors investigate the underlying reasons: technical, batch effects ...etc. that could explain such a disparity within the female donors?

The authors provide in the present version different new statistical analyses that added statistical significance to the data interpretation.

Lines 423 – 425: Which studies show that members of Proteobacteria are commonly considered to be environmental noise rather than true skin residents?

The authors did not understand the question and how they have to modify the text.

Lines 208-209: Authors claim that males harbor more bacteria as their read numbers are significantly higher (!). There is a confusion here between the bacterial biomass i.e. the total collection of bacterial material in a given sample, and the number of reads or sampling depth obtained through PCR and sequencing techniques. When pooled into a library for sequencing, samples need to be equimolar. If samples are sequenced with different starting concentrations, this will bias the subsequent analysis and thus comparison as some will be underrepresented and others underrepresented. In the sequencing run, sampling depth should be similar across samples.

To compare the total bacterial biomass across samples, techniques such cell sorting/counting, ddPCR with total bacteria primers on samples with identical concentrations can determine the bacterial biomass within a sample.

On the other hand, bacteria counts or relative abundances generated from high-throughput sequencing can be further corrected using the actual bacterial total biomass.

For further details, an example study:

Zemb O, Achard CS, Hamelin J, De Almeida ML, Gabinaud B, Cauquil L, Verschuren LMG, Godon JJ. Absolute quantitation of microbes using 16S rRNA gene metabarcoding: A rapid normalization of relative abundances by quantitative PCR targeting a 16S rRNA gene spike-in standard. Microbiologyopen. 2020 Mar;9(3):e977. doi: 10.1002/mbo3.977. Epub 2020 Jan 11. Erratum in: Microbiologyopen. 2022 Jun;11(3):e1305. PMID: 31927795; PMCID: PMC7066463.

There is a misunderstanding, the authors modify the text in order to better clarify the point.

Data not shown: Authors rely too often on undisclosed data to support claims. Data need to be reported, and discussed at least for review purposes.   

Reviewer 2 Report

In my opinion, the research is very preliminary. It is difficult to draw conclusions with such a small group and such a wide age range. An analysis of the age-dependent differences in the microbiota is extremely lacking. I recommend changing the nature of the work to preliminary studies

Author Response

Dr. Elisa Bona

University of Piemonte Orientale, DiSIT

Piazza San Eusebio 5

13100 Vercelli, Italy

e-mail: elisa.bona@uniupo.it

                                                                                                          Vercelli, 24th November 2022

Subject: Response to reviewers.

Dear Letitia Huang,

please find enclosed our revised manuscript entitled “Influence of Sex on the Microbiota of the Human Face” by Dr. Robert and coworkers.

The authors thank the reviewers for their comments in order to improve the manuscript. The authors hope that the applied modifications meet their expectations.

Best regards.

                                                                                                     Dr. Elisa Bona

Reviewer 2

In my opinion, the research is very preliminary. It is difficult to draw conclusions with such a small group and such a wide age range. An analysis of the age-dependent differences in the microbiota is extremely lacking. I recommend changing the nature of the work to preliminary studies

The author implemented the statistical analysis presentation and add the information in the paper.

Reviewer 3

  1. " ... other factors than can be ..." --> other factors that can be

DONE

  1. " ... NGS technologies ..." --> Next Generation Sequencing

The authors agree with the reviewer and they added the NGS definition in the text.

  1. 2.2 Sample collection

Provide detail regarding successive eNAt addition and the time and g-force of the centrifugation.

500-700uL of enat were added to the basket containing the swab head, and the 2mL tube+basket were centrifuged 1min at 10 000g. When the total volume of enat to be transferred was > 1mL, the flow-through was transferred in a clean 2mL tube, and the process above repeated until no enat would remain in the collection tube. All liquid handling was performed under a laliar folw hood.

  1. References

List up to 10 authors. If there are more than 10 authors, you may list all of 

them, or list the first 10 followed by et al.

The authors check the references and they responded to the reviewer suggestions.

  1. 2.1 Study design.  Presumably the 300 samples from which the 96 cheek samples were selected came from a previous study for which informed consent was obtained.  Please clarify this.

All participants enrolling in studies for Complife give their informed consent upon participating to said studies.

Reviewer 3 Report

1. " ... other factors than can be ..." --> other factors that can be

2. " ... NGS technologies ..." --> Next Generation Sequencing

3. 2.2 Sample collection

Provide detail regarduing successive eNAt addition and the time and

g-force of the centrifugation

4. References

List up to 10 authors. If there are more than 10 authors, you may list all of 

them, or list the first 10 followed by et al.

5.  2.1 Study design.  Presumably the 300 samples from which the 96 cheek

samples were selected came from a previous study for which informed 

consent was obtained.  Please clarify this.

Author Response

The results obtained in the present work, in the Author opinion, are not preliminary but supported by different kind of statistical approaches. Moreover, the results are in line with preliminary data reported in liberature.

Kind regards

Elisa Bona

Round 2

Reviewer 1 Report

Minor comments

- Abstract line 22: the samples were subject to amplicon sequencing after the 16S rRNA PCR, and not to genome sequencing- this is misleading

The authors agree with the reviewer and modify the text.

Indeed, thank you

- Why no analyses were performed on the sub-genus level? OTU, ASV

The bioinformatic analysis were performed at species level and obtained results were reported at genus level.

OK, thank you

- Table 3: The percentage indicate median or mean values?

Table 3 showed relative abundance values in males and females calculated by microbiome analyst selecting in taxa resolution the option “merging small taxa with counts < 10 based on median”.

Authors do not provide a clear answer. There were 48 females, and 48 males. These are the most abundant phyla, and only one percentage value per sex is reported. It remains unclear whether these are the median or the mean relative abundances per sex group. Such information should be in the table title/legend.

Major comments

Abstract line 16: "In fact, relatively little information is available about the composition of the healthy skin microbiota." Well this is no longer true, many studies described the skin microbiome in health, also in studies which investigated skin diseases such as atopic dermatitis, psoriasis and did recruit, and described the microbiome of healthy controls.

The authors are in partial agreement with the reviewer. Indeed, it is true that in the works that evaluate the variations in the diseased microbiota, the controls are described, but there are no specific insights related to the factors that modulate the healthy microbiota.

Incorrect. There are studies which investigated the factors influencing the healthy skin microbiome, including in Caucasian population, including the facial microbiome, and assessing sex influence.

Additionally, studies which recruited diseased and healthy controls, assessed the effect of internal, and external factors on the skin microbiome in the healthy and diseased individuals.

Here are a few studies:

Moitinho-Silva L, Boraczynski N, Emmert H, Baurecht H, Szymczak S, Schulz H, Haller D, Linseisen J, Gieger C, Peters A, Tittmann L, Lieb W, Bang C, Franke A, Rodriguez E, Weidinger S. Host traits, lifestyle and environment are associated with human skin bacteria. Br J Dermatol. 2021 Sep;185(3):573-584. doi: 10.1111/bjd.20072. Epub 2021 May 18. PMID: 33733457.

Park J, Schwardt NH, Jo JH, Zhang Z, Pillai V, Phang S, Brady SM, Portillo JA, MacGibeny MA, Liang H, Pensler M, Soldin SJ, Yanovski JA, Segre JA, Kong HH. Shifts in the Skin Bacterial and Fungal Communities of Healthy Children Transitioning through Puberty. J Invest Dermatol. 2022 Jan;142(1):212-219. doi: 10.1016/j.jid.2021.04.034. Epub 2021 Jul 10. PMID: 34252398; PMCID: PMC8688298.

In addition to the work of Hillebrand et a., 2021 and Oh et al, 2016 mentioned hereafter, the authors statement „In fact, relatively little information is available about the composition of the healthy skin microbiota“ is inaccuarte.

Abstract line 24: The authors claim to identify members of the facial skin microbiota that were not reported before in the literature; however this claim is not supported by the data presented, and authors do report in the discussion that identified taxa were described elsewhere.

Her is a study which examined skin microbiota from the cheek, and reports similar results:

Kim, HJ., Kim, J.J., Myeong, N.R. et al. Segregation of age-related skin microbiome characteristics by functionality. Sci Rep 9, 16748 (2019). https://doi.org/10.1038/s41598-019-53266-3

The author thank the reviewer for the correction and suggestion and modify the text accordingly.

The authors modified the claim„microbiota that were not reported“ to „microbiota that were rarely reported“…

Lines 90 – 94: Studies on the facial cutaneous microbiota are lacking in western populations: A recent and a landmark study did investigate the face microbiome including from cheek samples, and should be acknowledged.

Hillebrand GG, Dimitriu P, Malik K, Park Y, Qu D, Mohn WW, Kong R. Temporal Variation of the Facial Skin Microbiome: A 2-Year Longitudinal Study in Healthy Adults. Plast Reconstr Surg. 2021 Jan 1;147(1S-2):50S-61S. doi: 10.1097/PRS.0000000000007621. PMID: 33347075.

Oh J, Byrd AL, Park M; NISC Comparative Sequencing Program, Kong HH, Segre JA. Temporal Stability of the Human Skin Microbiome. Cell. 2016 May 5;165(4):854-66. doi: 10.1016/j.cell.2016.04.008. PMID: 27153496; PMCID: PMC4860256.

Done

These studies were indeed added by the authors

Study design, lines 104-106: More detailed metadata need to reported. For example what was the age distribution across the groups. Did the authors take into consideration further recruitment criteria including use of any medication – especially antibiotics – smoking status? Additional criteria such as life style (sport, ...), diet, all of which could influence the skin microbiome.

Smoking status was not considered, most of our volunteers don't smoke (about 85%).

The patients assuming antibiotic therapy were excluded from the study.

The authors do not address age distribution across individuals, even though they test for age influence on the facial microbiota. This information is still lacking. These is also very crucial to address any confounding effects of the tested factors, sex, age … etc.

Authors do provide an overall age range in the method section, and show age categories by sex in figure 1-c, however it is hard to see age distribution across sexes, and whether this is biased. A simple histogram or summary statistics of age would have been relevant.

Sample collection and DNA processing:

Blank samples: were negative collection controls (only buffers, sampling environment … etc.) included along with the skin samples during sampling? Same for DNA extraction negative controls? These controls are important for subsequent "decontamination" procedure

Blank samples were tested when establishing the general extraction procedure, but not further: since we process a high number of samples, it would be too expensive and unpractical. A negative control is included to each PCR round, and the absence of contamination is controlled via electrophoresis gel (along with the correct amplification of the bacterial DNA).

What was exactly tested on the blanck samples? Does this mean blanck samples were tested for contamination?, and found not contaminated (and how? Electrophersis?), and thus were not further included in the sequencing library? Authors do not provide a clear answer on what is meant by „blanck samples were tested, but not further“.

The argument that it would be too expensive to include negative controls is rather weak. Authors report data from 48 females, and 48 males. This is not a high number of samples for 16S rRNA amplicon sequencing. Second, Blank samples are usually collected for each round of data collection, and processing (e.g. DNA extraction round, sampling day), and not for each single sample collected. Third, clean PCR controls are actually required, and usually several PCR negative controls- and not just one- are added per round to control for the different barcoded primers.

Current quality standards for generating 16S rRNA amplicon sequencing data- especially for low biomass organs such as the skin, the lung- require assessing, and removal of contaminates using laboratory and analytical tools. Refer to this study:

Davis NM, Proctor DM, Holmes SP, Relman DA, Callahan BJ. Simple statistical identification and removal of contaminant sequences in marker-gene and metagenomics data. Microbiome. 2018 Dec 17;6(1):226. doi: 10.1186/s40168-018-0605-2. PMID: 30558668; PMCID: PMC6298009.

Library preparation: Positions and sequences of the primers, and PCR conditions need to reported. Were PCR negative controls (PCR reagents) checked? Dual or single primers barcoding was used?

The method proposed by Arrow is a kit that did not provide this information.

However, primer positions are crucial for data reproducibility and taxonomy comparisons.

What was the final concentration of the pool?

The final concentration of the pool was 5pM.

OK, thank you.

Were samples pooled to the same/equimolar concentration?

Yes, the samples were pooled at the same concentration.

OK, thank you.

How was the sequencing performed and the two sequencing kits used? The MiSeq® v2 kit is not sufficient to cover the entire V1-V3 length. Were positive controls (mock community of known composition) sequenced along?

The author add this information in the text.

Authors added a citation Torre et al., 2022. This study reports the same information present in the manuscript, thus it remains unclear how the authors sequenced the long V1-V3 16S rRNA region, using the v2 kit, which sequences up to 250 bp on each strand?

Microbiome analyses:

Line 136: Which RDP classifier released version was used? Taxonomy may differ across versions.

Done

Indeed, thank you

Lines 141-142:"Data was not filtered in order to retain singletons and doubletons from the occurrence table, which participate to a meaningful estimate" The state of the art approach – especially for low biomass organs- is to actually filter out not only singletons and doubletons, but also low abundant taxa, present at 1% or 2%. Numerous studies demonstrated that these low abundant taxa potentially generate misleading diversity profiles, an example study is the following:

Reitmeier, S., Hitch, T.C.A., Treichel, N. et al. Handling of spurious sequences affects the outcome of high-throughput 16S rRNA gene amplicon profiling. ISME COMMUN. 1, 31 (2021). https://doi.org/10.1038/s43705-021-00033-z

We agree with the reviewer and we added some more details in the text.

Authors indeed added a paragraph on data processing (lines 148 – 160). However, analysis pipeline remains confusing: line 157-158 “… remove features that are unlikely to be useful when modeling data”- this is unclear, based on what exact parameters sequences were filtered out and decided to not impact the downstream analyses? Also, “… low count and variance can be removed during filtration step” this is incomplete – were the sequences actually filtered out or not? The statement “can be filtered” is unclear.

Lines 142-144: "diversity should be assessed on the raw data since samples are not compared to each other but evaluated individually or between groups" Samples were indeed compared to each other- that´s the principle of beta diversity measures. Normalizing sampling depth across samples, should at least be evaluated on the performed analyses. Please have a look at this study:

Weiss S, Xu ZZ, Peddada S, Amir A, Bittinger K, Gonzalez A, Lozupone C, Zaneveld JR, Vázquez-Baeza Y, Birmingham A, Hyde ER, Knight R. Normalization and microbial differential abundance strategies depend upon data characteristics. Microbiome. 2017 Mar 3;5(1):27. doi: 10.1186/s40168-017-0237-y. PMID: 28253908; PMCID: PMC5335496.

OK

Indeed, authors removed the statement “samples are not compared to each other but evaluated individually or between groups” which was inaccurate.

Removal of contaminants: The skin harboring a low microbial biomass, influence of contaminants is crucial, and need to be addressed. Several analytical tools are available which rely on the negative controls collected during samples collection, and data generation. Such procedure is described in this recently published skin microbiota study:

Barnes CJ, Clausen ML, Asplund M, Rasmussen L, Olesen CM, Yüsel YT, Andersen PS, Litman T, Hansen AJ, Agner T. Temporal and Spatial Variation of the Skin-Associated Bacteria from Healthy Participants and Atopic Dermatitis Patients. mSphere. 2022 Feb 23;7(1):e0091721. doi: 10.1128/msphere.00917-21. Epub 2022 Feb 23. PMID: 35196118; PMCID: PMC8865923.

Upon establishing our analysis pipeline, both the repeatability and reproducibility of the results were assessed. The results were compared against the literature to control the absence of pollutants possibly introduced during sample manipulation.

Is the repeatability and reprehensibility assessment reported in the manuscript? Does it identify and remove environmental contaminants and/or cross-contamination among samples? More importantly, screening the literature to identify contaminants in the dataset is insufficient and not the current standard to handle contamination in microbiome data of low biomass organs. Please refer again to the mentioned current studies above.

Figure 2 shows a sub-group of females to be distinct from the remaining female and male donors, and appear to be driving the difference, and significance between the two groups. Did the authors investigate the underlying reasons: technical, batch effects ...etc. that could explain such a disparity within the female donors?

The authors provide in the present version different new statistical analyses that added statistical significance to the data interpretation.

Indeed, authors performed additional statistics to assess the effect of sex, and age on several community aspects. However, these analyses do not directly address the comment, which regards the clear disparity within female donors, even at phylum level. This is important because it might bias the statistics which compare females to males.

Lines 423 – 425: Which studies show that members of Proteobacteria are commonly considered to be environmental noise rather than true skin residents?

The authors did not understand the question and how they have to modify the text.

Authors claim- in lines 438 – 439 “Members of the Proteobacteria phylum are more commonly accepted as environmental bacteria” this claim needs to be precised supported by more citations.

Lines 208-209: Authors claim that males harbor more bacteria as their read numbers are significantly higher (!). There is a confusion here between the bacterial biomass i.e. the total collection of bacterial material in a given sample, and the number of reads or sampling depth obtained through PCR and sequencing techniques. When pooled into a library for sequencing, samples need to be equimolar. If samples are sequenced with different starting concentrations, this will bias the subsequent analysis and thus comparison as some will be underrepresented and others underrepresented. In the sequencing run, sampling depth should be similar across samples.

To compare the total bacterial biomass across samples, techniques such cell sorting/counting, ddPCR with total bacteria primers on samples with identical concentrations can determine the bacterial biomass within a sample.

On the other hand, bacteria counts or relative abundances generated from high-throughput sequencing can be further corrected using the actual bacterial total biomass.

For further details, an example study:

Zemb O, Achard CS, Hamelin J, De Almeida ML, Gabinaud B, Cauquil L, Verschuren LMG, Godon JJ. Absolute quantitation of microbes using 16S rRNA gene metabarcoding: A rapid normalization of relative abundances by quantitative PCR targeting a 16S rRNA gene spike-in standard. Microbiologyopen. 2020 Mar;9(3):e977. doi: 10.1002/mbo3.977. Epub 2020 Jan 11. Erratum in: Microbiologyopen. 2022 Jun;11(3):e1305. PMID: 31927795; PMCID: PMC7066463.

There is a misunderstanding, the authors modify the text in order to better clarify the point.

Indeed, authors removed the previous claim it appears that males present more bacteria on their cheeks, and their read numbers are also significantly higher” which was confusing bacterial biomass, and bacteria abundance.

Data not shown: Authors rely too often on undisclosed data to support claims. Data need to be reported, and discussed at least for review purposes.

Authors did not address this comment, there are seven mentions of “data not shown” in the revised manuscript. If data are not disclosed, they can not be reviewed.